# miR-34b/c-5p/CXCL10 Axis Induced by RSV Infection Mediates a Mechanism of Airway Hyperresponsive Diseases

**DOI:** 10.3390/biology12020317

**Published:** 2023-02-16

**Authors:** Dan Liu, Zhongxiang Tang, Ousman Bajinka, Pei Dai, Guojun Wu, Ling Qin, Yurong Tan

**Affiliations:** 1Department of Medical Microbiology, School of Basic Medical Sciences, Central South University, Changsha 410078, China; 2Department of Respiratory Medicine, National Key Clinical Specialty, Branch of National Clinical Research Center for Respiratory Disease, Central South University, Changsha 410078, China; 3National Clinical Research Center for Geriatric Disorders, Xiangya Hospital, Central South University, Changsha 410078, China; 4Hunan Provincial Clinical Research Center for Respiratory Diseases, Xiangya Hospital, Central South University, Changsha 410078, China; 5China-Africa Research Center of Infectious Diseases, Central South University, Changsha 410078, China

**Keywords:** respiratory syncytial virus, mRNA-miRNA, integrated analysis, miR-34b/c-5p, CXCL10, airway hyperresponsiveness

## Abstract

**Simple Summary:**

Airway hyperresponsive diseases (AHD), such as asthma and chronic obstructive pulmonary disease (COPD), are a serious public health burden worldwide. Studies have shown that viral infection plays an important role in asthma and COPD, especially RSV. In this study, we analyzed differentially expressed miRNAs (DEmiRs) in RSV-infected patients, asthma patients, and COPD patients by screening miRNA profiling from public datasets. Integrated analysis was then performed with mRNA datasets obtained from RSV-infected patients. We found that miR-34b/c-5p was downregulated. In vivo and in vitro experiments confirmed that decreased hsa-miR-34b/c-5p expression induced CXCL10 secretion and promoted THP-1 derived macrophages chemotaxis. In addition, miR-34c-5p can bind directly to CXCL10. This study provides new insights into the molecular mechanism of hsa-miR-34b/c-5p/CXCL10 in airway inflammation and AHR.

**Abstract:**

**Background:** RSV is closely correlated with post-infection airway hyperresponsive diseases (AHD), but the mechanism remains unclear. **Objective:** Due to the pivotal role of miRNAs in AHD, we analyzed the differentially expressed miRNAs (DEmiRs) in RSV-infected patients, asthma patients, and COPD patients from public datasets and explored the mechanisms of association between RSV and AHD. **Methods:** We obtained miRNA and mRNA databases of patients with RSV infection, as well as miRNA databases of asthma and COPD patients from the GEO database. Through integrated analysis, we screened DEmiRs and DEGs. Further analysis was carried out to obtain the hub genes through the analysis of biological pathways and enrichment pathways of DEGs targeted by DEmiRs and the construction of a protein-protein interaction (PPI) network. **Results:** The five differential molecules (miR-34b/c-5p, Cd14, Cxcl10, and Rhoh) were verified through in vivo experiments that had the same expression trend in the acute and chronic phases of RSV infection. Following infection of BEAS-2B cells with RSV, we confirmed that RSV infection down-regulated miR-34b/c-5p, and up-regulated the expression levels of CXCL10 and CD14. Furthermore, the results of the dual-luciferase reporter assay showed that CXCL10 was the target of hsa-miR-34c-5p. **Conclusions:** miR-34b/c-5p/CXCL10 axis mediates a mechanism of AHD.

## 1. Introduction 

Airway hyperresponsive diseases (AHD), such as asthma and chronic obstructive pulmonary disease (COPD), are a serious public health burden worldwide. Studies have shown that viral infection plays an important role in acute exacerbations of adult asthma and COPD [1,2,3]. RSV is an enveloped RNA virus with a single-stranded negative chain genome and a common pathogen that causes lung infections in infants and young children. Children who have experienced severe RSV infection during infancy leading to bronchiolitis have an increased likelihood of recurrent wheezing or asthma by age of 13 [4]. RSV aggravates post-infection asthma and COPD through multiple mechanisms and airway immune imbalance and involvement of the lung-brain axis are the main ideas [5,6,7]. At present, although there are many studies designed to study the mechanism of respiratory dysfunction caused by RSV infection [8,9], it remains unclear. 

In animal models, RSV infection can significantly increase the infiltration of neutrophils, eosinophils, and lymphocytes into the airways. It also induces the release of high levels of pro-inflammatory cytokines, including IL-1, IL-6, IL-12, IL-13, IFN-γ, and TNF-α, chemokines, such as CCL5, CXCL10, and CXCL11, and TNF-related apoptosis-inducing ligand (TRAIL) [10]. These contribute to airway inflammation, excessive mucus secretion, and AHR. During RSV infection, chemokines are key drivers of antiviral responses. They can interact with chemokine receptors to promote recruitment and activation of peripheral immune cells to resist viral infections [11]. However, excessive inflammatory response leads to enlarged lung injury, causing related respiratory diseases. Because a variety of chemokines exhibit higher expression levels in patients with airway diseases than in the control group, various small molecule antagonists of their receptors (such as CXCR3, CCR1, CCR4, and CCR5) have been developed to treat AHD [12].

MicroRNAs (miRNAs) are small non-coding single-stranded RNAs, usually composed of 18–25 nucleotides. They can induce the degradation of messenger RNA (mRNA) to inhibit the translation process of target genes, and negatively regulate gene expression at the post-transcriptional level. miRNAs are potential therapeutic targets by either inhibiting the viral genome or promoting host antiviral signaling. This is not only the case for RSV, but also for other respiratory viral infections [13]. Abnormal miRNA caused by RSV infection not only participate in the acute phase of inflammation, but also in the differentiation of immune cells and the expression of immune tolerance-related genes associated with AHR after viral infection [14]. Therefore, targeting specific miRNAs may be a potential prophylactic or therapeutic strategy for RSV-induced AHD. 

In this study, we analyzed differentially expressed miRNAs (DEmiRs) in RSV-infected patients, asthma patients, and COPD patients by screening miRNA profiling from public datasets. Integrated analysis was then performed with mRNA datasets obtained from RSV-infected patients. We found that miR-34b/c-5p was downregulated. In vivo and in vitro experiments were conducted to verify the expression of miR-34b/c-5p and its target genes, which may explain the inflammation mechanism of AHR. 

## 2. Materials and Methods 

### 2.1. Microarray Data 

The eight microarray expression profile datasets, including GSE62306, GSE33336, GSE142237, GSE69683, GSE103166, GSE56766, GSE106986, and GSE117827, were obtained from the Gene Expression Omnibus (GEO) database (https://www.ncbi.nlm.nih.gov/geo/; accessed on 12 November 2020) [15,16,17,18,19]. Datasets GSE62306, GSE33336, and GSE142237 are miRNA expression profiles. GSE62306 included nasal mucosa samples from 13 healthy children and 14 severely infected children with RSV using Agilent’s GPL19263 sequencing platform. GSE33336 contained 9 patients with mild COPD and 20 patients with moderate COPD from lung tissue using the GPL6955 sequencing platform of Agilent. GSE142237 had a total of 10 samples derived from bronchial epithelial cells, including 3 healthy controls and 7 asthmatics, using the GPL18058 sequencing platform of Exiqon miRCURY LNA microRNA array. GSE117827 was an mRNA expression profile containing 6 healthy control samples and 5 RSV-infected samples, derived from nasal mucosa samples from children after excluding other unrelated infections using the Affymetrix GPL23126 sequencing platform. GSE103166 and GSE106986 were used to analyze changes in hub gene expression in asthma and COPD airways, respectively, while GSE69683 and GSE56766 were used to analyze changes in hub gene expression in asthma and COPD blood, respectively. GSE103166 contains 106 samples using the Affymetrix GPL23961 sequencing platform. GSE106986 contains 19 samples using the Agilent GPL13497 sequencing platform. GSE69683 contains 498 samples using the Affymetrix GPL13158 sequencing platform. GSE56766 contains 206 samples using the Affymetrix GPL570 sequencing platform.

### 2.2. Differential miRNAs (DEmiRs) and Genes (DEGs) Screening 

R software (version 4.0.3; https://www.r-project.org/; accessed on 5 November 2020) and Bioconductor (http://www.bioconductor.org/; accessed on 5 November 2020) were used to process raw data. Sequencing platform files were used to convert probes and genes. When the genes had multiple probes, only the maximum probe corresponding data was taken. After the datasets were calibrated and normalized, the limma package was used for the screening [20].The R packages ggplot2 and heatmap were used for data visualization.

### 2.3. Prediction of miR-34b/c-5 Target Genes from DEGs 

The miRDB online tool (http://mirdb.org/; accessed on 17 November 2020) was used to obtain miR-34b/c-5p potential target genes. The predicted target genes of both miRNAs and up-regulated mRNAs in the GSE117827 dataset were analyzed by Venn, and significantly upregulated target genes were obtained after the intersection. 

### 2.4. Analysis of GO and KEGG Pathway 

Gene Ontology (GO) and Kyoto Encyclopedia of Genes and Genomes (KEGG) pathway enrichment analyses (https://amp.pharm.mssm.edu/Enrichr/; accessed on 14 November 2020) were used to demonstrate the functions and roles of target genes. 

### 2.5. Construction of PPI Network and Identification of Hub Genes 

The differentially expressed target genes were uploaded to the STRING’s online database (https://string-db.org/; accessed on 14 November 2020). Cytoscape software (version 3.7.1, Seattle, WA, USA) was then used to analyze PPI networks based on STRING results. Cytoscape’s Molecular Complex Detection (MCODE) plug-in was applied to find the modules of the entire network. The hub genes were identified using the Cytoscape software plug-in cytoHubba, including density of maximum neighbourhood component (DMNC) and maximal clique centrality (MCC) [21]. The top 10 genes with the highest connectivity were identified as key genes. 

### 2.6. Cell Culture and Viral Preparation 

Human lung epithelial BEAS-2B cells, Hela cells, and HEK293T cells were cultured in DMEM plus 10% fetal bovine serum (FBS) at 37 °C and 5% CO_2_. RSV (type A2) was stored by the Department of Medical Microbiology, Central South University. RSV was propagated in Hela cells with DMEM containing 3% FBS and viral titers were determined by plaque assay. 

### 2.7. Transfection of miRNA Mimics and RSV Infection 

Synthetic miR-34b/c-5p mimics (50 nmol/L) or negative control (RiboBIO, Guangzhou, China) were transfected into BEAS-2B cells following riboFECT CP Transfection Kit (RiboBIO, C10511-05, Guangzhou, China) for 24 h. Subsequently, cells were infected with RSV at MOI = 1 for 48 h. 

### 2.8. Dual-Luciferase Reporter Assay 

Simply, 3′-UTR fragments of CXCL10 containing miR-34c binding site were amplified by Nanjing Genscript Biological Technology Co., Ltd. and cloned into pmirGLO vector (Promega, Madison, WI, USA). Then, wild type CXCL10 or mutant CXCL10 3′-UTR (CXCL10-WT or CXCL10-MUT) was constructed. Next, the constructed plasmids were co-transfected with miR-34c-5p mimics or mimics-NC into 293T cells using Lipofectamine 2000 (Invitrogen, Waltham, MA, USA) according to the manufacturer’s instructions. After 48 h of transfection, luciferase activity was measured by the dual-luciferase reporter system (Promega, USA). 

### 2.9. Animal Models of RSV-Induced AHR

Animal models of AHR induced by RSV were established according to the previous method [22]. Briefly, 6-8 weeks BALB/c female mice weighing 16–20 g (purchased from Hunan Tianqin Biological Technology Co., Ltd., Changsha, China) were kept in a pathogen-free environment. The mice were randomly divided into the control group (n = 12) and the RSV group (n = 12). After anaesthetizing the mice with isoflurane, 5 × 10^6^ pfu RSV in 100 μL, was intranasally inoculated. For mock infections, mice were given an equivalent volume of sterile PBS. The airway resistance of mice was tested on day 7 and 28 post-infection (6 mice from each group). The mice were then sacrificed, and their lungs were taken for follow-up studies. RSV infection was verified by indirect immunofluorescence (IFA) with RSV major surface glycoprotein G monoclonal antibody (Bioss, bs-1264R, Beijing, China) as the primary antibody and CY3-conjugated antibody (Boster, BA1032, Wuhan, China) as the secondary antibody. 

### 2.10. Real-Time RT-PCR 

Total RNA was extracted from lung tissue or cells using TRIzol reagent (Takara, Osaka, Japan). Primers were synthesized by Sangon Biotech Co., Ltd. (Shanghai, China), and the gene names and primer sequences of the genes were shown in Appendix A. Each sample was reverse transcribed into cDNA using RR036A PrimeScript RT Master Mix (Perfect Real Time) (Takara, Japan). For miRNA cDNA synthesis, use RR037A PrimeScript RT Master Mix (Perfect Real Time) (Takara, Japan) with miRNA-specific stem-loop RT primers. The cDNA was then synthesized by reverse transcription and amplified using 2× SYBR Green qPCR Master Mix (Bimake, Houston, TX, USA) according to the manufacturer’s instructions. qPCR was performed at 95 °C for 3 min, 40 cycles of 95 °C for 15 s, 55 °C for 30 s, and 72 °C for 30 s. U6 was the internal reference for miR-34b/c, and GAPDH was the internal reference for other target genes. Relative mRNA expression levels were calculated using the 2^−ΔΔCT^ method.

### 2.11. Measurement of Airway Responsiveness to Methacholine 

The airway resistance of the mice was measured on the 7th and 28th day of RSV infection by the Buxco pulmonary function testing system (Buxco, Sharon, Wilmington, NC, USA). After anesthetizing the mice with sodium pentobarbital (60 mg/kg), the airway was separated and tracheal intubation was performed, and then the mice were mechanically ventilated with a small animal ventilator at a tidal volume of 10 mL/kg, and a frequency of 120 breaths/min. Then, mice inhaled 10 μL of methacholine (0.00, 6.25, 12.5, 25, and 50 mg/mL), and airway resistance values (Rn, cm H_2_O, s/mL) were obtained by measuring airway flow and pressure. Methacholine was inhaled by atomizing methacholine after endotracheal intubation. Each concentration was separated by two minutes and the resistance value collected within four minutes of initial ventilation without methacholine challenge was used as the baseline value.

### 2.12. Hematoxylin-Eosin Staining 

Left lung tissue taken from each animal was fixed in a 10% formaldehyde solution for 24 h, then embedded in paraffin and cut into 5 μm sections. Hematoxylin-eosin staining was performed according to routine experimental procedures. Then, histomorphological changes were observed under an optical microscope. 

### 2.13. Immunofluorescence 

Lung tissue sections were fixed with 95% ethanol and 0.1% Triton-X100. Samples were then blocked with normal goat serum for 20 min and incubated overnight with RSV major surface glycoprotein G monoclonal antibody (Bioss, bs-1264R, Beijing, China) rabbit primary antibody at 4 °C. Then, the samples were incubated with a CY3-conjugated goat anti-rabbit secondary antibody (Boster, BA1032, Wuhan, China) at room temperature for 1 h. After being counterstained with DAPI for 10 min, samples were observed under a light microscope at high magnification (200×) (Leica, Wetzlar, German). 

### 2.14. ELISA 

CXCL10 levels in culture supernatants of BEAS-2B cells were measured using ELISA kits (Jianglaibio, Shanghai, China) according to the manufacturer’s protocol. 

### 2.15. PMA Induced THP-1 Monocytes to Differentiate into Macrophages

Phorbol 12-myristate 13-acetate (PMA), a protein kinase C inducer, induced cell differentiation into macrophages.THP-1 cells were cultured in 1640 medium containing 10% FBS, 1% penicillin-streptomycin, and 0.05 mM β-mercaptoethanol (Sigma, St. Louis, MO, USA), and then passed on to the third generation. After 1 mg PMA (Sigma, St. Louis, USA) was dissolved in DMSO, it was divided and diluted to a final concentration of 100 nM. THP-1 cells were cultured for 48 h. Morphological changes in cells were observed under a microscope.

### 2.16. Chemotaxis Assay

BEAS-2B cells with or without RSV and/or miR-34b/c-5p were placed in the upper chamber of the Transwell chamber for 48 h at 37 °C in an atmosphere containing 5% CO_2_. Then, the culture medium was discarded, and a new medium was added. THP-1-derived macrophages (2 × 10^4^ cells) were then placed in a culture medium (600 μL) containing 10% FBS in the lower chamber. The cells were then incubated for 24 h at 37 °C in an atmosphere containing 5% CO_2_. The cells on the lower surface were then stained with 4% crystal violet. Afterwards, they were incubated at room temperature for 15 min and then washed three times with PBS. Chemotactic cells were observed and counted under an inverted microscope (Leica, Wetzlar, Germany). Each group was randomly photographed in 5 fields, and cell counts were performed on each field, then averaged. 

### 2.17. Statistical Analysis 

GraphPad Prism version 7.0 was used for statistical analysis, and the data were expressed as mean ± standard deviation. T-test was used for comparison between two groups. One-way ANOVA was used for comparison among multiple groups, and LSD was used for post hoc test. *p* < 0.05 was considered significant. 

## 3. Results 

### 3.1. miR-34b/c-5p Expression Was Downregulated in Patients with RSV Infection Asthma, and COPD

To understand which miRNAs are disregulated in RSV infection, moderate COPD, and asthma, a differential analysis of microarray data using three different datasets was performed. The raw data of each dataset were normalized following data pretreatment. A total of 40 DEmiRs (26 up-regulated and 14 down-regulated) were identified in GSE62306, while four DEmiRs were identified in GSE33336, of which two were up-regulated and two were down-regulated. A total of 178 DEmiRs were identified in GSE142237, of which 82 were up-regulated and 96 down-regulated. Volcano plots (Figure 1A–C) show the DEmiRs. A total of two down-regulated miRNAs, including miR-34b/c-5p (0 up-regulated miRNAs), were finally screened in the 3 miRNA microarray datasets following Venn analysis (Figure 1D). Figure 1E–G demonstrated miR-34b/c-5p expression in patients with RSV infection asthma, and COPD, respectively. 

### 3.2. CXCL10 Is the Most Important Downstream Gene of miR-34b/c-5p

To find the target genes of miR-34b/c-5p, integrated analysis was then performed with mRNA datasets obtained from RSV-infected patients. For microarray data GSE117827, DEGs were reflected in the volcano plot (Figure 2A). A total of 3379 DEGs (497 up-regulated and 2882 down-regulated) were identified in RSV-infected patients. Principal component analysis (PCA) showed that the DEGs were effective in distinguishing two groups. To further explore DEGs related to the mechanism of RSV-induced airway inflammation and AHR, we obtained 1231 potential target genes for hsa-miR-34b/c-5p through the miRDB online tool. The predicted target genes were then compared with the DEGs in microarray data GSE117827 and 23 up-regulated intersect genes were screened (Figure 2B). 

Further analysis was carried out to obtain the hub genes through the analysis of biological pathways and enrichment pathways of DEGs targeted by DEmiRs and the construction of a protein-protein interaction (PPI) network. GO enrichment analysis showed that the DEGs targeted by hsa-miR-34b/c-5p were mainly enriched in biological processes, including cytokine-mediated signaling pathways, and cellular response to bacterial molecules. Moreover, they enriched in molecular functions consisting of histone threonine kinase activity. In addition, the DEGs targeted by miR-34b/c-5p were mostly located in the recycling endosome and trans-Golgi network transport vesicle (Figure 2C). The KEGG enrichment analysis revealed that the DEGs targeted by hsa-miR-34b/c-5p were predominantly enriched in inflammatory pathways, such as NF-kappa B signaling and chemokine signaling (Figure 2D). 

The PPI network was constructed from an online STRING tool (minimum required interaction score: 0.9) and the result was visualized with Cytoscape software [23]. The 23 intersecting genes were uploaded and analyzed by STRING and Cytoscape software. Then, 10 hub genes, including LCP1, CCR1, CXCL10, CD14, ZEB1, PRKCB, PZRY14, IRAK2, RHOH, and LILRA1, were selected from cytoHubba analysis (Figure 2E). The color depth of the figure represents the weight of these genes in this network, of which CXCL10 has the highest weight and ZEB1 has the lowest weight.

### 3.3. miR-34b/b-5 P Downregulation Was Accompanied by CXCL10 Activation in Mouse RSV-Infected Models

Next, expression of miR-34b/c and hub genes in the lung tissue were validated in the mouse model of AHR induced by RSV infection. RSV (type A2) was propagated in Hela cells and the viral titer was determined using plaque assay. Then, the mouse model of AHR induced by RSV infection was constructed and RSV infection in the lungs of mice was verified using IFA and real-time PCR (Figure 3A,B). Results showed significant red fluorescence (RSV major surface glycoprotein G) in the RSV group compared to the control group (Figure 3A). The expression of viral F protein was also significantly increased (Figure 3B). To confirm the promotion effect of RSV infection on airway inflammation, the pathological examination was analyzed by H&E staining. Compared to control mice, peribronchial and perivascular infiltrating inflammatory cells were increased in RSV-infected mice (Figure 3C). Rn values after aerosolizing different concentrations of methacholine were significantly increased in the RSV-infected group compared to those in the control group (Figure 3D). During acute RSV infection period (7 d), AHR and airway inflammation in RSV-infected mice were the most severe. Over time, these symptoms were gradually alleviated during the chronic phase of infection (28 d). 

Then, mouse miR-34b/c-5p (mmu-miR-34b/c-5p) and hub genes were detected by qRT-PCR. The results showed that mmu-miR-34b/c-5p were downregulated in the lung tissues of RSV-infected mice (7 and 28 d) compared to control mice (Figure 3E). Furthermore, we found that the levels of *Cxcl10*, *Cd14*, and *Rhoh* in lung tissues were significantly increased on day 7 of RSV infection, as compared to those in the control group (LILRA1 is not expressed in mice) (Figure 3F). On day 28, only levels of *Cxcl10*, *Cd14*, and *Rhoh* were detected as higher than in control mice (Figure 3G). 

### 3.4. CXCL10 Was Consistently Overexpressed in Lung Tissue, While CD14 Was Consistently Over-Expressed in Blood Samples from RSV Infection, Asthma, and COPD

To further confirm cellular localization of hub genes in lung tissue, we then obtained publicly available GEO data and performed a synergistic analysis of expression of 10 hub genes in the lung tissues from three diseases (Figure 4A–C). Differential analysis showed that all 10 genes were upregulated after RSV infection. CXCL10 was upregulated in asthma and LCP1, CCR1, PRKCB, IRAK2, LILRA1, and ZEB1 were significantly upregulated in COPD patients (Figure 4D). These results showed that CXCL10 was consistently overexpressed in lung tissues of RSV infection, asthma, and COPD.

We also performed a synergistic analysis of Hub genes in blood samples from three diseases and showed that 10 hub genes were also consistently upregulated in blood samples (Appendix A), particularly CD14 expression (Appendix A). We then validated the expression of 10 hub genes in blood samples from the RSV-infected mice model, where *Cxcl10*, *Cd14*, *P2ry14*, and *Irak2* were significantly upregulated after seven days, but no gene was upregulated at 28 days (Appendix A). 

We also studied the expression distribution of CXCL10, RHOH, and CD14 in different cell lines using the online expression Atlas database and found that CXCL10 is highly expressed in pulmonary endothelial cells, epithelial cells, and stem cells. Moreover, RHOH is mainly expressed in CD4^+^ T cells and CD8^+^ T cells, while CD14 is mainly expressed in monocytes and macrophages and various types of inflammatory cells (Appendix A).

### 3.5. miR-34b/c-5p Inhibits CXCL10 Expression through Direct Interaction with CXCL10 mRNA

Considering the role of epithelial cells in AHD, we further applied epithelial cells to verify miR-34b/c-5p target genes. We infected BEAS-2B cells with RSV for 48 h and confirmed infection using IFA and qRT-PCR. Compared to uninfected cells, red fluorescence was seen in BEAS-2B cells (Figure 5A,B). Then, we tested the mRNA levels of hsa-miR-34b/c-5p and hub genes (CXCL10, CD14, and RHOH) using qRT-PCR. In RSV-infected BEAS-2B cells, hsa-miR-34b/c-5p expression levels were downregulated (Figure 5C), and CXCL10 and CD14 levels were upregulated (Figure 5D), which were consistent with the trend of animal experiments. However, RHOH showed no significant difference between the two groups (Figure 5D). To further estimate the function of hsa-miR-34b/c-5p in the expression of CXCL10 and CD14 after RSV infection, hsa-miR-34b/c-5p mimics were transfected into BEAS-2B cells prior to RSV infection and transfection efficiencies were shown in Figure 5E. As shown in Figure 5F, the mRNA levels of CXCL10 decreased when transfected with hsa-miR-34b/c-5p mimics prior to RSV infection, while CD14 levels showed no difference. Furthermore, protein expression levels of CXCL10 were also significantly reduced compared to the RSV+ miR-NC group (Figure 5G). We then predicted that CXCL10 could bind to hsa-miR-34b/c-5p using bioinformatics analysis and found that CXCL10 was a potential target gene for hsa-miR-34c-5p, but not hsa-miR-34b-5p (Figure 5H). Luciferase assays were performed to detect the relationship between CXCL10 and hsa-miR-34c-5p. As illustrated in Figure 5G, overexpression of hsa-miR-34c-5p significantly inhibited the luciferase activity of the CXCL10-WT reporter vector but not the CXCL10-MUT reporter vector. Taken together, our findings suggest that hsa-miR-34c-5p may inhibit CXCL10 expression by directly targeting it. The sequences of miR-34b and miR-34c had only a few base differences, and the predicted binding sites of miR-34b and CXCL10 mRNA are completely consistent with those of miR-34c and CXCL10 mRNA, so it is possible that miR-34b also decreases CXCL10 mRNA through direct interaction.

### 3.6. miR-34b/c-5p Inhibited Macrophages Chemotaxis to RSV-Infected BEAS-2B Cells

Next, we investigated whether downregulation of miR-34b/c-5p in BEAS-2B cells induced by RSV infection affected macrophages chemotaxis in the co-culture system. After treating THP-1 cells with PMA for 48 h, cell morphology changed significantly. The cells were changed from the original suspension to adhesion, and the morphology was from round to protruding (Figure 6A). A co-culture system of BEAS-2B cells and THP-1-derived macrophages was established using the Transwell method, and then the chemotaxis of macrophages to BEAS-2B cells was detected. Compared with the control group, the chemotaxis of macrophages in the RSV infection group was significantly enhanced, while the chemotaxis of macrophages in the RSV+hsa-miR-34b-5p-mimics group and RSV+hsa-miR-34c-5p-mimics group were significantly lower than those in the RSV+miR-NC group (Figure 6B). In addition, RSV infection promoted CXCL 10 secretion in the co-culture system, which was downregulated in miR-34b/c-5p treated BEAS-2B cells (Figure 6C). The results showed that RSV-infected BEAS-2B cells could promote the chemotaxis of THP-1-derived macrophages to BEAS-2B cells, and this process was regulated by hsa-miR-34b/c-5p.

## 4. Discussion 

RSV can cause severe lower respiratory tract infections, especially in infants, immunocompromised individuals, and the elderly. Persistent RSV infection can lead to subsequent asthma. In addition, 64% of COPD exacerbations are caused by respiratory infections, such as RV, influenza, and RSV, affecting the bronchial epithelium. Therefore, the inflammation may be the common basis for RSV infection, asthma, and COPD. 

In this study, a joint analysis of miRNA-mRNA expression profiles of three disease models was performed to understand the mechanism of RSV infection affecting the occurrence and development of AHR at the systemic level. miR-34b/c-5p expression was downregulated in patients with RSV infection asthma, and COPD with striking consistency. We then constructed a mouse model of RSV-induced AHR and found that miR-34b/c-5p expression in the lungs of RSV-infected mice was downregulated, but the selected hub genes, Cxcl10, Cd14, and Rhoh, were highly expressed during RSV infection. 

The miR-34 family is a group of highly conserved miRNAs, including miR-34a, miR-34b, and miR-34c. The expression of miR-34a is highest in the brain, while in the lung, miR-34b and miR-34c are mainly expressed [24]. The miR-34 family is very important in regulating virus–host interaction by participating in immune response and virus replication. They can be used as biomarkers for viral infection [25]. Some studies have shown that miR-34 is closely related to the occurrence and development of lung diseases such as lung cancer and asthma [26,27,28]. Solberg et al. found that miR-34s (miR-34c, miR-34b) expression in bronchial epithelial cells in asthma patients was significantly down-regulated [29]. Yin et al. reported that miR-34/449 may promote airway inflammation and fibrosis by modulating IGFBP-3-mediated autophagy activation [26]. Hai-Xiang et al. reported that miR34c-5p targeting CCL22 plays a protective role in chronic obstructive pulmonary disease [30]. In addition, miR-34b/c-5p can downregulate mucus secretion by regulating MUC5AC and FGFR1 [31,32]. This shows that miR-34b-5p and miR-34c-5p may serve as therapeutic drug targets for reducing inflammation caused by immune cell infiltration into the lung in AHD. Through miRNA screening of RSV infection, asthma, and COPD, we found that miR-34b/c-5p was significantly down-regulated in these three disease models, which to some extent showed that there was a certain commonality among the three disease models. Therefore, focusing on miR-34b/c-5p will help us understand the inflammation mechanism involved in RSV infection, asthma, or COPD. However, the role of miR-34b/c in RSV infection has not yet been clarified. 

Among the target genes of miR-34b/c, Cd14, Cxcl10, and Ras homologous gene family H (Rhoh) were continuously increased in terms of expression at day 7 and 28 in this study. We found in the dataset that CXCL10 was highly expressed in pulmonary epithelial cells and endothelial cells. Viral infection may lead to the release of CXCL10 in the bronchial epithelium, which activates mast cells and allows them to migrate to airway smooth muscles to exacerbate airway inflammation and bronchoconstriction in asthma [33]. Moreover, lung epithelial cells are also the main site of RSV infection, so we chose lung epithelial cells to continue our experiments. We also found in the dataset that CD14 was highly expressed in blood cells. CD14 is a positive biomarker of macrophages, and chemokines are the main mediator of macrophage homing. Xuan et al. have found that CXCL10 can induce the chemotaxis of M1 macrophages [34]. Beena Puthothu’s research has shown that the toll like receptor 4 (TLR4)/CD14 complex is particularly important for initiating an innate immune response to RSV [35]. Studies have shown that the number of macrophages in the lungs of COPD patients increased, and lung macrophages play a key role in initiating and continuing the chronic inflammatory response. We hypothesized that CXCL10 secretion was a chemotactic factor for macrophages. Using Transwell’s method, we confirmed that RSV induced macrophages chemotaxis by inhibiting hsa-miR-34b/c-5p expression in BEAS-2B cells. 

However, the study still has many questions to answer. First, we screened 23 miR-34b/c-5p target genes in human studies. Only a few genes have been tested at the animal level, probably because of the difference between laboratory animals and humans. Secondly, because miR-34b/c-5p target genes are distributed in different tissues, e.g., CD14 is mainly distributed in macrophages and RHOH is mainly distributed in T cells, whether they are miR-34b/c-5p target molecules remains to be further investigated. Third, the RSV infection, COPD, and asthma datasets seem quite small. Whether RSV infection, COPD, and asthma function via the miR-34b/c-5p/CXCL10 axis needs to be further investigated by collecting more clinical samples. We hope to further investigate the effects of miR-34b/c-5p/CXCL10 on the occurrence and development of AHR and their predictive and therapeutic values in AHR. 

In conclusion, this study used bioinformatics to jointly analyze the expression profiles of three disease models including RSV infection, COPD, and asthma, and we also found the commonality among the three disease models. Five common differential molecules (hsa-miR-34b/c-5p, CD14, CXCL10, and RHOH) were validated, which provided effective targets for clinical diagnosis and treatment of AHR. In vitro experiments confirmed that decreased expression of hsa-miR-34b/c-5p induced CXCL10 secretion and promoted THP-1 derived macrophage chemotaxis. In addition, miR-34c-5p can bind directly to CXCL10. Based on the above results, we elucidate a possible new mechanism of RSV inducing and exacerbating AHD through the hsa-miR-34b/c-5p/CXCL10 axis. Similarly, this study provides new insights into the molecular mechanism of hsa-miR-34b/c-5p/CXCL10 in airway inflammation and AHR. 

## Figures and Tables

**Figure 1 biology-12-00317-f001:**
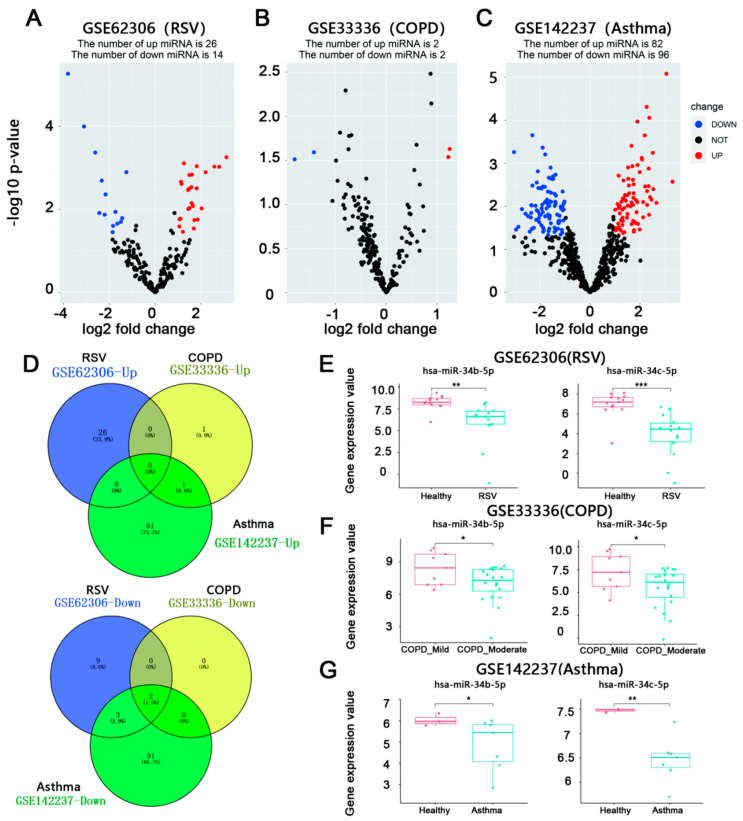
**Aberrantly expressed miRNA molecules in RSV infection, asthma or COPD were assessed by volcanic plot.** (**A**–**C**), volcano map for DEmiRs in GSE62306, GSE33336, and GSE142237. Blue spots represented underexpressed miRNA molecules, while red spots represented overexpressed miRNA molecules. Black spots represent non-differentially expressed molecules. A total of 26 up-regulated and 14 down-regulated DEmiRs were identified in GSE62306, while 2 up-regulated and 2 down-regulated DEmiRs were in GSE33336. A total of 82 up-regulated and 96 down-regulated DEmiRs were identified in GSE142237. (**D**), Venn plots revealed the same miRNAs with significant changes were found in patients with RSV infection, asthma, and moderate COPD.miR-34b/c-5p were downregulated and no upregulated miRNA were found. (**E**–**G**), miR-34b/c-5p expression levels in GSE62306, GSE33336 and GSE142237. The values were expressed as log2 fold change. miR-34b/c-5p expression was downregulated in RSV infection, asthma, or COPD (* *p* < 0.05, ** *p* < 0.01 and *** *p* < 0.001).

**Figure 2 biology-12-00317-f002:**
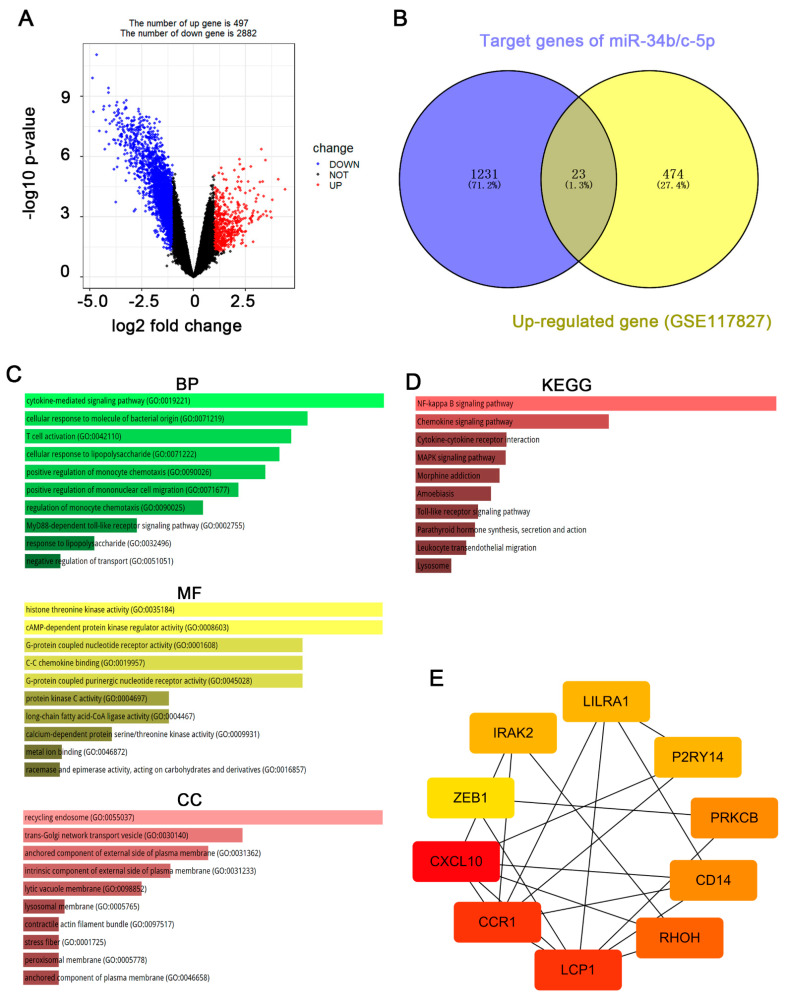
**DEGs in RSV−infected patients and screening for differentially expressed target genes.** (**A**), volcano map for 3379 different mRNA expressions. Dots in red and blue indicate high and low mRNA expression in patients with RSV infection. (**B**), Venn plot showed the distribution of predicted target genes and upregulated in DEGs. (**C**), Results of GO analysis revealed the relationship between DEGs targeted by DEmiRs and functional pathways. The length of the bar represented −log10 for the P value. (**D**), KEGG pathway of DEGs targeted by DEmiRs. The length of the bar represented −log10 for the *p* value. (**E**), STRING PPI network data were further analyzed by Cytoscape, and hub gene identification was performed by cytoHubba. The color depth of the figure represents the weight of these genes in this network, of which CXCL10 has the highest weight and ZEB1 has the lowest weight. GO, Gene Ontology; BP, Biological Processes; CC, Cellular Components; MF, Molecular Functions; KEGG, Kyoto Encyclopedia of Genes and Genomes. PPI, Protein-Protein Interaction.

**Figure 3 biology-12-00317-f003:**
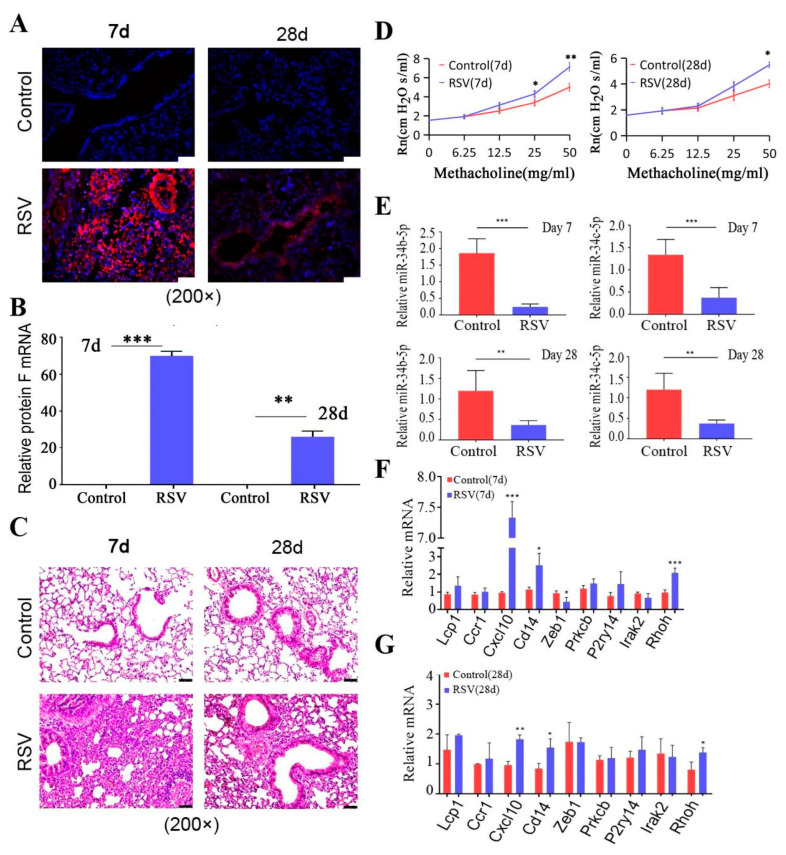
**Expression of miR−34b/c and hub genes in the lung tissue of RSV−induced AHR.** (**A**), RSV infection in the lungs was verified using IFA (magnification 200×). Significant red fluorescence was observed in the RSV group. (**B**), RSV infection in lung tissues was assayed using qRT-PCR. The viral protein F was significantly increased. (**C**), Pathological examination was analyzed by H&E staining (magnification 200×). Compared to control mice, peribronchial and perivascular infiltrating inflammatory cells were increased in RSV-infected mice. (**D**), AHR to methacholine was evaluated by a Buxco pulmonary function testing system. Rn values after aerosolizing different concentrations of methacholine were significantly increased in the RSV-infected group. (**E**–**G**), detection of mRNA levels of miR-34b/c-5p and hub genes in lung tissues using qRT-PCR (n = 6) miR-34b/c-5p were downregulated in lung tissue of RSV-infected mice (7 and 28 d) compared to control mice (Figure 3E). The levels of *Cxcl10, Cd14,* and *Rhoh* in lung tissue were significantly increased on day 7 and 28 of RSV. (* *p* < 0.05, ** *p* < 0.01 and *** *p* < 0.001 vs. Control).

**Figure 4 biology-12-00317-f004:**
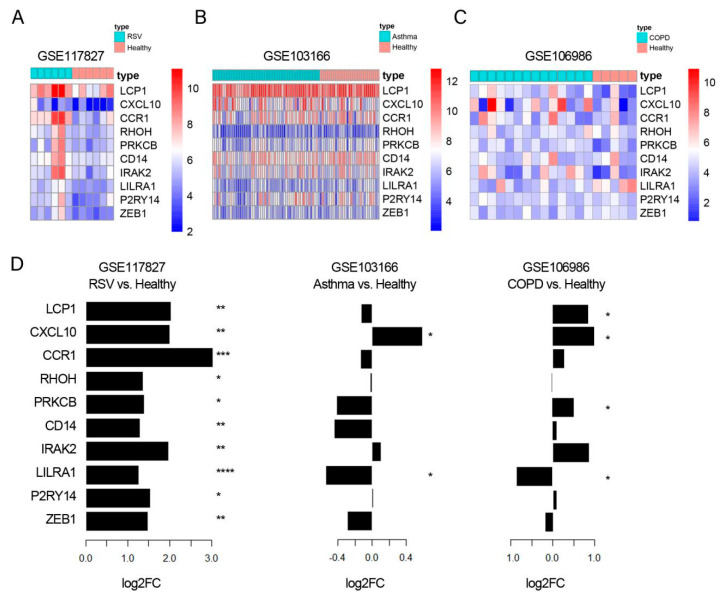
**Synergistic analysis of expression of 10 hub genes in the lung tissues from three diseases.** (**A**–**C**), expression profiles of 10 hub genes in the lung of RSV infection, asthma, and COPD. Each column showed a single sample, each row represented a marker gene, and its expression value was normalized among 50 genes. (**D**), barplot displays differential analysis of expression of 10 hub genes in the lung of RSV infection, asthma, and COPD (* *p* < 0.05, ** *p* < 0.01, *** *p* < 0.001, and **** *p* < 0.0001 vs. healthy), log2 fold change (LogFC).

**Figure 5 biology-12-00317-f005:**
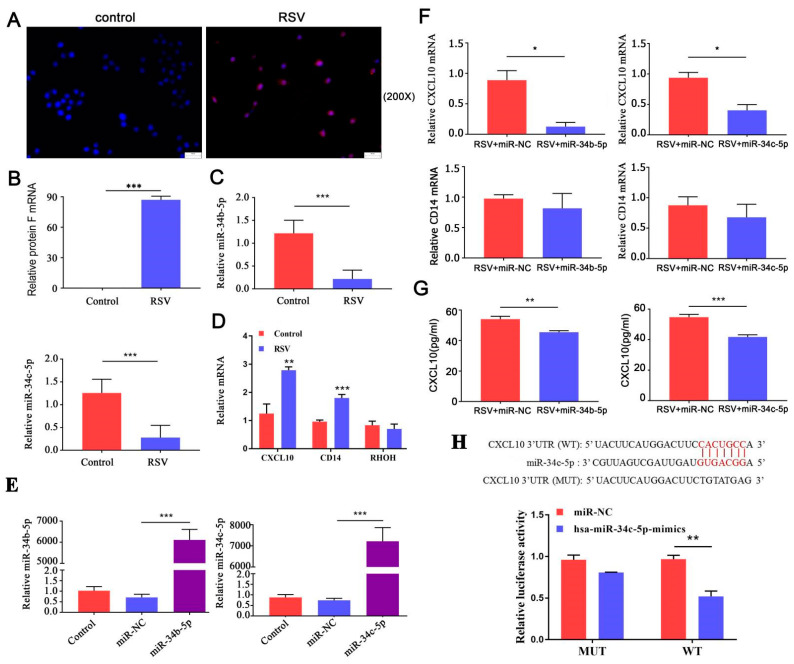
**Downregulated miR−34b/c−5p induced CXCL10 expression in RSV−infected BEAS−2B cells (n = 5).** (**A**), RSV infection in BEAS-2B cells was verified by indirect IFA with RSV G protein monoclonal antibody (magnification 200×). Red fluorescence was seen in BEAS-2B cells. (**B**), RSV infection in BEAS-2B cells was verified by qRT-PCR. (**C**,**D**), the expression levels of miR-34b/c-5p and hub genes (CXCL10, CD14, and RHOH) in BEAS-2B cells (infected with RSV for 48 h) were assayed using qRT-PCR. In RSV-infected BEAS-2B cells, hsa-miR-34b/c-5p expression levels were downregulated, and CXCL10 and CD14 levels were upregulated. (**E**), BEAS-2B cells treated with miR-34b/c-5p mimics (mimics-miR-34b/c) or negative control mimics (miR-NC), and qRT-PCR were used to detect transfection efficiencies. (**F**), miR-34b/c-5p mimics or negative control mimics were transfected into BEAS-2B cells prior to 48 h of RSV infection, using qRT-PCR to detect the mRNA levels of CXCL10 and CD14. The mRNA levels of CXCL10 were decreased when transfection with hsa-miR-34b/c-5p mimics before RSV infection, while CD14 levels showed no difference. (**G**), ELISA was used to detect CXCL10 protein level. CXCL10 protein expression levels were also significantly reduced compared to the RSV+ miR-NC group. (**H**), miR-34c binding sites within CXCL10 predicted by bioinformatics tools. Luciferase reporter assay was performed to validate the relationship between CXCL10 and miR-34c-5p. Over-expression of hsa-miR-34c-5p significantly inhibited the luciferase activity of the CXCL10-WT reporter vector but not the CXCL10-MUT reporter vector. (* *p* < 0.05, ** *p* < 0.01, and *** *p* < 0.001).

**Figure 6 biology-12-00317-f006:**
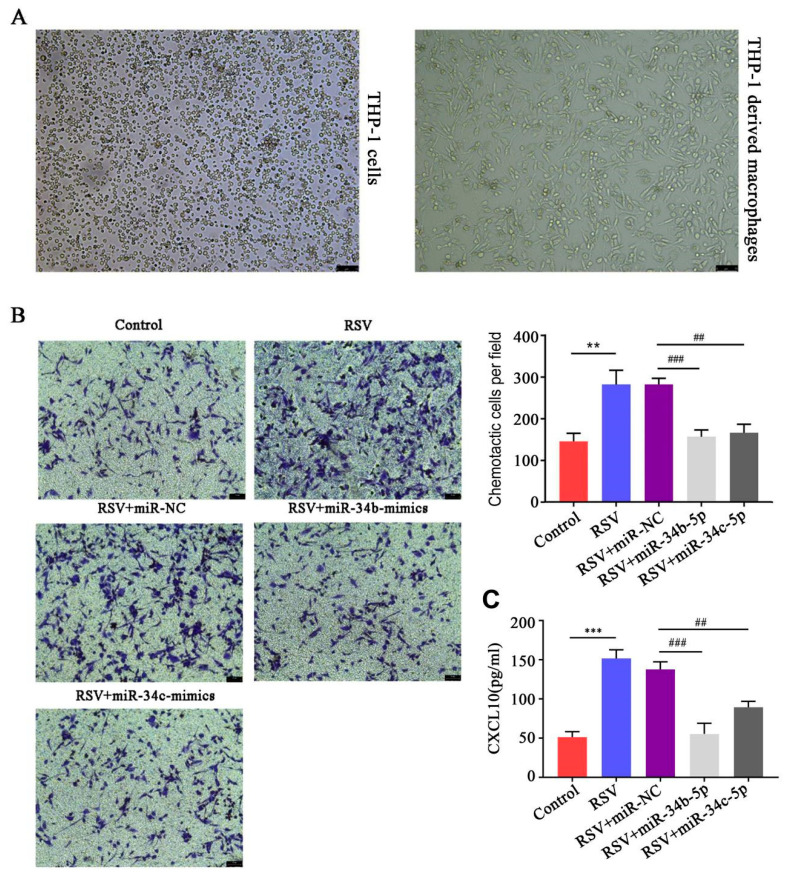
**miR−34b/c−5p inhibited the chemotaxis of co-cultured THP−1−derived macrophages with RSV−infected BEAS−2B cells (n = 5).** (**A**). Changes in cellular morphology of PMA-induced THP-1 monocytes (100×) (Left: THP-1 cells; Right: PMA-induced THP-1 derived macrophages). After treating THP-1 cells with PMA for 48 h, the cells changed from the original suspension to adhesion, and the morphology was from round to protruding. (**B**). RSV infection promoted macrophages chemotaxis in the co-culture system, which was aboragated by miR-34b/c-5p. The quantification diagram is shown on the right (** *p* < 0.01 vs. Control; ## *p* < 0.01 and ### *p* < 0.001 vs. RSV+miR-NC). (**C**). RSV infection promoted CXCL 10 secretion in the co-culture system, which was aboragated by miR-34b/c-5p. (*** *p* < 0.001 vs. Control; ## *p* < 0.01 and ### *p* < 0.001 vs. RSV+ miR-NC).

## Data Availability

Data sets used and/or analyzed during the current study are available from the corresponding author upon reasonable request.

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
