# Peer review of "miR-34b/c-5p/CXCL10 Axis Induced by RSV Infection Mediates a Mechanism of Airway Hyperresponsive Diseases"

_biology, 2023, doi:10.3390/biology12020317_

Round 1

Reviewer 1 Report (Previous Reviewer 1)

The Authors substantially improved their manuscript, therefore, I recommend the paper to be accepted for publication.

The only minor comments I would recommend the Authors to address are related to the figures:

1) Figure 3D - I would recommend the Authors to use "red" labelling consistently for control and "blue" for RSV group.

2) Figure 4A - I would recommend the use of consistent labelling here as well, using "red" for healthy and "blue" for RSV samples.

Author Response

The only minor comments I would recommend the Authors to address are related to the figures:

  • Figure 3D - I would recommend the Authors to use "red" labelling consistently for control and "blue" for RSV group.

Response: That's good advice. A uniform grouping of colors will help readers better understand our study, which we have revised as you suggested.

2) Figure 4A - I would recommend the use of consistent labelling here as well, using "red" for healthy and "blue" for RSV samples.

Response: We have modified it in accordance with your suggestion.

Reviewer 2 Report (New Reviewer)

I have carefully read the paper by the authors. It is we ll written and well designed study.

The methods should be more briefly written.

The results should be clearly summarized.

The conclusion needs more focus on the topic.

There are different colors of the font. It makes it hard to read. It has to be read by a native speaker

Author Response

I have carefully read the paper by the authors. It is we ll written and well designed study.

The methods should be more briefly written.

Response: Indeed, the method section is long. But because of many steps we took in this study, and because of the detailed description of the methods section, we were able to help people understand them effectively. We made some simplifications in the revised version.

The results should be clearly summarized.

Response:This opinion is of considerable importance. We gave a brief summary as needed. For example:

3.1 miR-34b/c-5p expression were downregulated in patients with RSV infection asthma, and COPD

3.2 CXCL10 is the most important downstream gene of miR-34b/c-5p

3.3 CXCL10 expression was activated by down-regulating miR-34b/c-5p in mouse RSV-infected models

3.4 CXCL10 was consistently over-expressed in the respiratory tract, while CD14 was consistently over-expressed in the blood of RSV infection, asthma and COPD

3.5 miR-34b/c-5p inhibits CXCL10 expression through direct interaction with CXCL10 mRNA

The conclusion needs more focus on the topic.

Response:Indee. In response to this, we have deepened our theme at the conclusion and changed it to “In conclusion, this study used bioinformatics to jointly analyze the expression profiles of three disease models including RSV infection, COPD, and asthma, and we also found the commonality among the three disease models. Five common differential molecules (hsa-miR-34b/c-5p, CD14, CXCL10, and RHOH) were validated, which provided effective targets for clinical diagnosis and treatment of AHR. In vitro experiments confirmed that decreased expression of hsa-miR-34b/c-5p induced CXCL10 secretion and promoted THP-1 derived macrophages chemotaxis. In addition, miR-34c-5p can bind directly to CXCL10. Based on the above results, we elucidate the new mechanism of RSV inducing and exacerbating AHD through hsa-miR-34b/c-5p/CXCL10 axis. Similarly, this study provides new insights into the molecular mechanism of hsa-miR-34b/c-5p/CXCL10 in airway inflammation and AHR. ”

There are different colors of the font. It makes it hard to read. It has to be read by a native speaker

Response:This is because the article is in the process of revision. We have preserved and reddened the modified marks. The text color will be unified at the end of the subsequent revision.

Reviewer 3 Report (New Reviewer)

Author Response

Major comments
• I do not agree with the major conclusion drawn in this paper. Concluding that miR-34b/c-5p/CXCL10 axis mediates an important mechanism of AHD is not justified based on your data. You can only state that there is an association between AHD (or rather inflammation) and this axis, no causality. It seems probable to me that this axis is activated upon all inflammation situations in the lung (also upon other respiratory infections). So this might not be linked to AHD but just to lung inflammation (which is present in RSV, COPD and asthma). To prove that the miR-34b/c-5p/CXCL10 axis is an important mechanism, you would need to make a knock-out model and test if you still have AHR.

Response: Thank you for your good suggestion. First of all, I would like to thank you for yout hard work in reviewing our study.Chronic airway inflammation is a common pathological feature of AHD. RSV infection, asthma and COPD all belong to AHD. In our study, we found that miR-34b/C -5p was significantly up-regulated in all of them. The association between AHD (or inflammation) and miR-34b/C -5 P/CXCL10 axis suggests that miR-34b/C -5 p/CXCL10 axis is associated with AHD through inflammation, which is consistent with numerous studies. The knock-out animal model will be the next step in our research.

  • It is not clear why you chose these specific datasets. Especially comparison betweenmoderate and mild COPD seems not relevant. Unless the moderate COPD patientsexperienced an exacerbation due to RSV? Also, for the asthma patient dataset, is there a link with RSV? Did these people have severe RSV as an infant which predisposed them for asthma? Also for the RSV dataset (GSE62306), at what point were the samples taken during their RSV infection? Did these children develop AHD?

Response: This study was conducted in 2019. At that time, there were few public data sets on miRNA in asthma, COPD, and RSV, and only these few data sets with the same sampling location were selected. Therefore, these specific data sets are selected for analysis. Comparison between moderate and mild COPD, because this data set explored the association between RSV infection and exacerbation of COPD. Other samples in public databases were not given clinical information about RSV infection. In order to make the study rigorous, we used the data set of the original literature cited.

  • The introduction does not clearly state the rationale of your research. Clearly point out whatthe aim is of the research and why you use this methodology.
    • I do not understand your AHR mouse model. To have proper AHR induction after RSVinfection you should first sensitize with an allergen (such as in Peebles et al, Journal of medical virology 1999).

Response: The AHR mouse model used in this experiment is an effective animal model of airway hyperresponsiveness verified by our laboratory. We carefully read the literature that you provided, and although the literature indicates that RSV infection alone did not cause significant airway hyperresponsiveness to methacholine, our study does the opposite. The animal model of persistent RSV infection was established by using a large dose of virus in our group. The airway resistance was significantly increased in both acute and persistent infection stages. Junyan Han et al's report published in 2010 in Am J Respir Crit Care Med also suggested that RSV infection alone could establish an animal model of airway hyperresponsiveness. So we don't think allergen induction is necessary.

  • Fig 4c and D. How is the significance determined? There does not seem to be a relevantdifference in the GSE57148 dataset between healthy and copd. The Log2FC scale bar is verysmall for this dataset. The differences might be significant but not relevant! This analysis seems to me over-powered.

Response: This is indeed a problem, for which we re-selected the sequencing data set in the GEO database and re-analyzed it with GSE106986, and the upregulation of CXCL10 is quite significant as shown in figure 4D. The reason why some data sets are obviously different, we speculate that it may be due to the individual differences of samples, sequencing batch, sequencing platform, sequencing operation and so on. The results of the bioinformatics analysis were verified experimentally in the present study.

  • How does miR-34b works (it also reduces the CXCL10 mRNA) if it is not by direct interaction.There is no discussion on this.

Response: When we compared the sequences of miR-34b and miR-34c, we found that the sequences of miR-34b and miR-34c had only a few base differences, and the predicted binding sites of miR-34b and CXCL10 mRNA are completely consistent with those of miR-34c and CXCL10 mRNA, so it is possible that miR-34b and miR-34c also decrease CXCL10 mRNA through direct interaction. In addition, we searched the pubmed database with mir-34b and CXCL10 as keywords and no relevant literature was reported, which we explained it in the Results section. However, whether miR-34b directly leads to the down-regulation of CXCL10 mRNA is still a question worthy of further investigation.

Minor comments
• Introduction: RSV does not induce COPD. RSV can only lead to an exacerbation of COPD. It also does not induce asthma. A severe infection can predispose for the development of asthma.

Response: Thanks for your good suggestion. Indeed, RSV does not induce COPD and asthma. But, it plays an important role in acute exacerbations of COPD and asthma. We revised the article to make the description more precise.

  • English can be improved (eg sentence in abstract: ‘the five differential molecules werescreened through animal experiments that had…’ is difficult to understand). Anotherexample in the discussion: ‘inflammatory mechanism may be the common basis for rsv infection, asthma and copd.’ At the basis of RSV is not an inflammatory mechanism but a viral infection.

Response:Thanks very much. In fact, we have also noticed some problems in English writing, which often lead to confusion and ambiguity in our description. and we have again revised the accuracy of the article. We revised the above sentences.

  • Introduce your results sections. Start each result section with an introductory sentence.

Response: We made some modifications as below:

3.1 miR-34b/c-5p expression were downregulated in patients with RSV infection asthma, and COPD

3.2 CXCL10 is the most important downstream gene of miR-34b/c-5p

3.3 CXCL10 expression was activated by down-regulating miR-34b/c-5p in mouse RSV-infected models

3.4 CXCL10 was consistently over-expressed in the respiratory tract, while CD14 was consistently over-expressed in the blood of RSV infection, asthma and COPD

3.5 miR-34b/c-5p inhibits CXCL10 expression through direct interaction with CXCL10 mRNA

  • Figure 1: The numbers of the datasets are not informative. Put more informative titles soyou can understand the figure without going to the materials and methods.

Response: Thank you for your comments. For the quantitative information on the dataset in Figure 1, we have described it in detail in Materials and Methods section. We have added some simple conclusions in the Figure legends.

  • Point out more clearly in the text why you focus on miR-34b/c-5p

 Response: We focused on mir-34b/c -5p because bioinformatics analysis showed that it showed striking consistency in airway hyperresponsiveness diseases. So we focused on its function. We made a supplement in the article.

  • You mention Fig.1C when you mean Figure 1E-G.

Response:This is an editing error, which we have corrected in the text.
• I do not understand the ‘However’ on page 9 (‘However in lung tissues during chronic infection,…) ,there is no contrast?

Response: We revised the sentences.
• Include size of scale bar (fig 3)

Response:The scale here represents a microscope magnified 200 times.

Round 2

Reviewer 3 Report (New Reviewer)

Title: How do you know it’s an important mechanism? It could be a redundant mechanism.

Introduction

In vivo and in vitro experiments were conducted to verify the expression of miR-34b/c-5p and its target genes, which may explain the mechanism and predict the occurrence and development of AHR.

I would phrase more carefully. There is no indication that expression of this miRNA can predict the occurrence and development of AHR. It’s merely an indication that inflammation is ongoing.

Results section

·         3.1. miR-34b/c-5p expression were downregulated in patients with RSV infection asthma, and COPD  à ‘WAS downregulated’.

·         The article is still lacking some introduction into each paragraph.  An extra title does not solve this. Eg for the first paragraph: To understand which miRNAs are upregulated in RSV infection, moderate COPD and asthma, a differential analysis of microarray data using three different datasets was performed.

·         Please adapt figure 1A-D so it is clear what you are looking at (just adding titles: RSV, COPD, Asthma would help). The numbers of the datasets are not descriptive.

·         Still do not understand the ‘however’ in ‘Furthermore, we found that the levels of Cxcl10, Cd14, and Rhoh in lung tissues were significantly increased on day 7 of RSV infection, as compared to those in the control group (LILRA1 is not expressed in mice) (Fig. 3F). However, in lung tissues during chronic infection, only levels of Cxcl10, Cd14, and Rhoh were detected higher than in control mice (Fig. 3G).    At both timepoints Cxcl10, Cd14 and Rhoh are increased, do you mean that there is a more limited increase at 28d? Please make this clear.

·         3.4. CXCL10 was consistently over-expressed in the respiratory tract, while CD14 was consistently over-expressed in the blood of RSV infection, asthma and COPD   Strange sentence (blood of RSV infection?)

·         Figure 4, the sample names under the expression profiles have no use as they are not readable for figure b.  As for the asthma data, a 0.4 log2 fold change (so 1.3 fold increase) does not seem highly relevant. I would point this out in the text. For the COPD data, I do not really understand, you did the re-analysis with a smaller sample size? So now there is a bigger difference. Seems like cherry-picking to me..   Also the RSV data set seems quite small to be able to omit individual differences between samples. I would be careful drawing conclusions from this data because either your data set is too small or the differences between the healthy and sick group are too small. I would consider removing the data, or at least pointing out the weakness (high inter sample variation, limited fold increase in the asthma and the COPD group) in evaluating this data. 

·         so it is possible that miR-34b also decrease CXCL10 mRNA through direct interaction.’ Typo: decreases.  Why did you not include miR-34b in the luciferase assay?

·         Figure 5, you forgot the ‘H’ in the caption

·         ‘Transwell’ is not a person. You describe it like it is a person (‘Transwell’s results, Transwell’s method’), rather than a type of cell culture plate.

Discussion

·         miR-34b/c-5p expression were’ expression is singular à not ‘were’ but ‘was’. Please have the manuscript read by a native English speaker to improve the language overall.

·         and RSV infection was an important factor inducing and aggravating asthma and COPD.’ RSV does not aggravate asthma but a severe RSV infection predisposes for the development of asthma. This is not the same.

·         Therefore, focusing on miR-34b/c-5p will help us understand the mechanism involved in RSV-induced exacerbation of asthma and COPD.’ This is a wrong conclusion, indeed this pathway is activated in the three conditions but this does not neem that this is involved in the exacerbation of asthma and copd. (again RSV does not exacerbate asthma). You can only say that this is a common mechanism in the three conditions, you cannot conclude a causal relationship.  

·          ‘We found that CXCL10 was highly expressed in …’ You did not observe this yourself, it was found in a dataset. The same for CD14.

·         Difference between animals and humans? Humans are animals, I presume you mean laboratory animals.

·         Based on the above results, we elucidate the new mechanism of RSV inducing and exacerbating AHD through hsa-miR-34b/c-5p/CXCL10 axis.’ It’s not because this pathway is activated in the three models that this is the mechanism of AHD. (‘a possible new mechanism’).

Author Response

This manuscript is a resubmission of an earlier submission. The following is a list of the peer review reports and author responses from that submission.

Round 1

Reviewer 1 Report

The Authors conducted an integrative analyis on the potential involvement of miR-34b/c-5p in RSV infection-induced CXCL10 upregulation-associated airway hyperresponsiveness using bioinformatics, in vitro and in vivo methods. The results provide valuable information that could contribute to the better understanding of the molecular pathogenesis and thus identifying novel targets for drug development.

The paper could be accepted for publication after addressing the following questions:

Materials and Methods:

-        - Why did the Authors not elaborate on the datasets GSE103166, GSE57148 used for the synergistic analysis of the expression of 10 hub genes? What about datasets GSE69683, GSE56766 – what were they used for? Apart from the 2.1. point I have not come across with these throughout the results.

-        -  Please enlist at least the names of target genes used for real-time PCR analysis in the methods section.

-        -  How was inhalation of metacholin performed during ventilation of intubated animals? Please include the protocol for metacholin challenge (data acquisition time, recovery time between challenges) in the methods part.

-         - STRING vs STING abbreviations are used inconsistently.

Result:

-         - Fig. 1 panel labeling is inconsistent with the figure legends. Fig 1D legends states that Venn plots revealed 2 up-regulated miRNAs, which again is inconsistent with the figure and text.

-         - Fig 2 panel labeling is inconsistent with legends.

-         - The Authors state RSV-infected mice exhibited increased peribronchial and perivascular infiltration of inflammatory cells, but the results are not convincing based on only 1 representative picture of each group. It would be more informative to include at least semi-quantitative scoring for the relevant parameter(s).

-         - Fig 3D demonstrates Rn changes after metacholin challange. I would recommend to use the same range on y-axis for better representation of time-dependent Rn changes. Please do not use linear x-axis, but rather show only the proper metacholin concentrations used on the x-axis, it would be more informative.

-         - Why are the control values of real-time PCR analyses not 1.0 in all cases?

-         - The Authors state that „levels of Cxcl10, Cd14, Zeb1 and Rhoh in lung tissues were significantly increased on day 7 of RSV infection”, which is again controversial based on Fig 3F.

-         - Please use consistent labeling for „Healthy” data on Fig 4.

-         - The Authors state that „The results revealed a significant upregulation of CD14 expression in all three diseases (Figure 4D)”, which looks inconsistent with the Figure, based on which in Asthma and COPS patients CD14 expression showed downregulation, although not significant.

-         - The Authors then elaborate on the results of the 10 hub genes in the blood of RSV-infected mice and refer to Fig 4E, which is clearly not the data the Authors refer to as.

-         - AHR and AHD are used may times throughout the text, although AHR is not once elaborated, one can only guess it stands for airway hyperresponsiveness and used as a synonim for AHD.

Author Response

Introduction

Some grammar and language correction needed.

Insufficient referencing in the introduction. Article needs referencing for background information. Further, there are two statements saying “studies have shown… “ without any reference.

Answer: We have added reference 1-3,8,9 in the Introduction Section.

A short statement about whether the miRNAs you have implicated in RSV disease are encoded by the host or the virus would be useful for clarity.

Answer: There are no reports that RSV itself encodes a miRNA. The transcript-altered miRNA is encoded by the host itself, and miR-34b/c-5p expression is altered after RSV infection.

Materials and methods.

Section 2.1

First 3 datasets are described as miRNA expression profiles. Does that mean that the other 5 used are just mRNA? This is unclear. Several datasets are undescribed in the text.

Answer: It is true that the description of the public datasets in this study is too simplistic, and we described these in details, see section 2.1. In addition, the number of samples in some datasets is very small, which is indeed a regret of this study. As more studies pay more attention to the direction of this topic, a large sample of research will appear. Similarly, our subsequent research will also be further conducted in this direction to make up for this regret. Regarding the COPD dataset, there are 9 cases of mild COPD and 20 cases of moderate COPD, because the public dataset does not currently have a miRNA sequencing dataset on normal and COPD, but COPD is a progressive disease, and the difference between mild and moderate COPD can reflect the logic of COPD development, so this dataset is appropriate to apply here.

2.11 Pentobarbital dose is unclear. Also, if it is 181 mg/kg, this is a very high dose that can affect immunological markers.

Answer: The dose of sodium pentobarbital is 60 mg/kg, we have revised it.

Results

3.1 Data pre-treatment is not described or referenced.

Answer: We described the processing of the dataset in detail and also refer to the R package in the Methods. See 2.2.

3.1 The numbers don’t add up. GSE142237 had a total of 109 ‘differentially expressed miRNAs’, with 82 up regulated and 96 down regulated. This makes a total of 178. Similarly, 10 DEmiRs in GSE33336, 2 up, 2 down = 4.

Answer: Yes, it was an editing error. We have corrected it and highlighted it in red (page 5).
Figure legend states the figure shows “A-C, Volcano map for DEmiRs in GSE62306, GSE33336, and GSE142237”, but these data sets are described in the materials and methods as having a mix of healthy and sick individuals. It is not stated whether the up or down regulation of gene expression was in the healthy or the sick subpopulations of the miRNA datasets. This is critical for drawing any conclusions about the role these genes play during infection.

Answer: The legend here only needs to describe the volcano map, and there is no room to elaborate what genes are up-regulated or down-regulated in the healthy population or the diseased population in the miRNA dataset. We have shown the list of miR34b and miR34c whose expression was down-regulated in the diseased subpopulation in supplementary Table 2 of the supplementary materials.

Figure 1A, X axis title typo “log2 fold change”

Answer: We have made corrections.

Figure 1B, numbers are unclear in Venn diagrams.

Answer: In order to improve the readability of the pictures, we have re-edited the pictures.

Figure 1C. The colour of the box plots for the healthier population and less healthy populations switches. Should keep consistent.

Answer: The color of the picture has been consistent as required.

Figure 2A axis title, again should be log2 fold change

Answer: Corrections have been made.

Figure 2C. Higher resolution image or different colour/layout needed to make text legible. Also, an axis title and a description in the figure legend of what the bars represent to anyone who is not familiar with a ‘GO analysis’. Description of BP, MF, CC and KEGG should be with the 2C section of the figure legend.

Answer: We re-edited the pictures to achieve higher clarity and better readability. The description of the BP, MF, CC and KEGG has been located in the figure legends.

2D. Figure legends need more information. Is this for the network diagram? What do the colours mean?

Answer: Yes, this is the constructed protein-protein interaction network. Ten hub genes are obtained through cytoHubba plug-in in Cytoscape. The color depth of the figure represents the weight of these genes in this network, among which CXCL10 has the highest weight and ZEB1 has the lowest weight.

2E. Where is 2E?

Answer: The annotation of the picture has been re-added.

3.3 Reference to figure 3A about pathological changes containing syncytia. There is no visible syncytia (characterised by multinucleated cells) in that image. All microscopy images in that figure need higher resolution. If referring to specific features in a histopathology image, arrows or some kind of marker are required.

Answer: We deleted the images and added the detection of qRT-PCR for RSV RNA.

“Results showed significant red fluorescence”. This is not informative. Describe what the IFA was targeting so that the reader doesn’t have to refer to the materials and methods.

Answer: We have added it. It is RSV major surface glycoprotein G.

The term ‘mmu-miR-34b/c-5p’ is introduced with no description of what ‘mmu’ means.

Answer: 'mmu' means mouse. Human miR usually starts with 'hsa', while mouse miR starts with 'mmu'. However, the corresponding miR of human and mouse is the same. However, this experiment is conducted in mice, so 'mmu' needs to be added before miR. We added it in the paper.

Figure 3 figure legend needs much more detail. Reader needs to be able to interpret the data without reading the rest of the paper.

Answer: We added the details.

 Figure 4. B and C, the writing under the graph is overlapping and unreadable. Remove or describe by a different method.

Answer: Figure 4. B and C, the writing under the graph is the stacking of sample names due to the huge number of samples, which is unavoidable when using R language tools to process data.We deleted them.

Figure 4D. logFC on x axis is not explained. Log fold change? Figure legends throughout this paper need much more detail.

Answer: Yes, Log2FC means log2 fold change. We explained this in the description of the legend. Modified parts are marked in red.

 Figure 5. Axis titles and microscopy are difficult to see. Better image quality needed. Figure legend needs information about the infection conditions for B and C (48 hours?).

Answer: The BEAS-2B cells were infected with RSV for 48h. We have added it in Figure legend.

Figure 5D was just confirming transfection efficiency? Were the cells treated or washed to remove the miRNA in solution? Some details are needed.

Answer: Yes,Figure 5D was just confirming transfection efficiency. Before collecting the cells, we have washed the cells with PBS 2-3 times.

5E. Can’t review as I can’t read the text on the figure.

Answer: We've re-edited the image and raised the resolution of the image to 600dpi,

5G. If the y axis is ‘relative luciferase activity’, is it relative to the miR-NC value? If this is the case, the mean value for miR-NC luciferase activity should be at 1.0, instead of the error bar being at 1.0.

Answer: When calculating miR-NC luciferase, we first calculated the average value of miR-NC luciferase, and then compared the actual value of a single miR-34b/c-5p luciferase with the average value of miR-NC luciferase. Finally, the luciferase values with error line was obtained. We believe that this calculation method is more accurate. We use this approach in our later calculations.

3.6. It is claimed that miR-34b/c-5p are implicated in chemotaxis but there is insufficient evidence to make this claim. What is provided is transwell migration, microscopy and cell rounding.  

Answer: We have added a detailed figure legends to demonstrate the question.

Figure 6A, it is not shown or described in the legend which is treated with the miRNA. This is critical for interpreting the results.

Answer: In Figure 6A, the cells were not treated with the miRNA. It only showed the THP-1 derived macrophages induced by PMA (Left:THP-1 cellsï¼›Right:THP-1 derived macrophages induced by PMA). We have added information in the legends.

Figure 6B. The is no way that the images provided can be used to draw the brief conclusions (with no details) made in the figure legend.

Answer:We added a detailed description.

Further, the transwell method is not described at all in either the relevant results section or the materials and methods. I suggest this section is removed or that the claims are softened substantially. It is a strong claim to suggest that miR-34b/c is involved in chemotaxis, given that this miRNA has been quite extensively studied in cancer and other contexts and has never before been linked to chemotaxis. Also, transwell figure in figure 6 has no description?

Answer: We added the methods description in Materials and Methods section. This part of the study is very weak, but the CD14 molecule is a biomarker of macrophages, which is also the target of miR-34b/c and we feel that the miR-34b/c-5 p/CXCL10/CD14 axis also exists.

Discussion.

The discussion is mostly more literature review instead of an analysis and interpretation of the results. miR-34b/c is described as regulating mucus which was ‘consistent with our results’, but no work has been done on mucus or mucus-related activity in this paper.

Answer: We deleted the sentence”consistent with our results” and rewrote the discussion.

Again, the conclusions about chemotaxis are unjustified and there is no explanation of this ‘Transwell’ method on which so many conclusions are drawn.

Answer:  In figure 6, we add some evidence of chemotaxis.

Reviewer 2 Report

Although RSV is a very common virus, it can sometimes cause severe problems in low-birthweight infants with immature lung function. In this paper, the authors focused on mRNA expression, which plays an important role in the mid(early)- to late-stage of RSV infection. Furthermore, the authors attempted to obtain more specific results by showing miRNA-mRNA candidates in the three major lung diseases using human cells derived from patients and RSV-infected mouse models.

To enhance the value of this paper, we request that you supplemental explanation on  some of the results and respond to the following comments.

1.           I would like to ask about the relationship between other lung diseases, asthma and COPD, and post-RSV infection: did the specimens of the 29 patients (9 with mild COPD, 20 with moderate COPD) have any symptoms or history of RSV infection in the past few months?

2.           Since the severity of COPD patients is described in Materials and Methods, the severity of asthma patients should also be described. Asthma severity is described in the latest asthma guidelines published by The Global Initiative for Asthma (GINA). Also, as you are aware, treatment for asthma patients varies by age, both for adults and children. It would be more helpful if you could indicate the age of each sample or the average of the patients.

3.           You should also indicate 7 samples from asthmatics and a recent history of RSV infection; all samples from RSV patients are from children. How did you exclude samples from unrelated infections to assess mRNA expression of GSE117827, RSV and other respiratory infections may coexist. If you have information on the samples you used, please tell us how you ruled out the possibility of co-infections.

4.           Why did you decide to measure blood samples as well as tissue and mucosa samples? Also, please provide some representative publications using CD-14 to evaluate RSV infections and other respiratory diseases.

5.           This paper provides a comparison of key mRNA expression in human samples and mouse models. The mouse model also allows for direct measurement of lung function and histology. The results of these experiments make this paper more informative. On the other hand, experiments using patient specimens are also very simple, so why did you add experiments using the mouse RSV infection model? Could you reiterate the significance of adding the results of the mouse RSV infection model to this paper and the limitations of this study?

In Abstract: 
Please put a right parenthesis “)” just after “AHD” in the first sentence of the “Abstract: Background:”.

Author Response

Comments and Suggestions for Authors

The Authors conducted an integrative analyis on the potential involvement of miR-34b/c-5p in RSV infection-induced CXCL10 upregulation-associated airway hyperresponsiveness using bioinformatics, in vitro and in vivo methods. The results provide valuable information that could contribute to the better understanding of the molecular pathogenesis and thus identifying novel targets for drug development.

The paper could be accepted for publication after addressing the following questions:

Materials and Methods:

-        - Why did the Authors not elaborate on the datasets GSE103166, GSE57148 used for the synergistic analysis of the expression of 10 hub genes? What about datasets GSE69683, GSE56766 – what were they used for? Apart from the 2.1. point I have not come across with these throughout the results.

Answer: Data sets GSE103166 and GSE57148 were used to analyze the expression of 10 hub genes in respiratory tract of different diseases. GSE103166 was for asthma patients, and GSE57148 was for COPD patients, as described in Figure 4. The corresponding data sets GSE69683 and GSE56766 were used to analyze the expression of 10 hub genes in the blood of different diseases, among which GSE69683 was for patients with asthma and GSE56766 was for patients with COPD, as described in supplementary Figure 1. We described this in more details in Materials and methods in Part 2.1.

-        -  Please enlist at least the names of target genes used for real-time PCR analysis in the methods section.

Answer:The primer sequences of all measured genes are listed in the supplementary table 1, and the corresponding gene list is also listed. Therefore, we modified the expression in the manuscript to "and the gene names and primer sequences of the genes were shown in Supplemental Table 1."

-        -  How was inhalation of metacholin performed during ventilation of intubated animals? Please include the protocol for metacholin challenge (data acquisition time, recovery time between challenges) in the methods part.

Answer:Yes. This was described in the manuscript as follows: metacholine was inhaled by atomizing metacholine after endotracheal intubation. Each concentration was separated by two minutes and the resistance value collected within four minutes of initial ventilation without metacholine challenge was used as the base value.

-         - STRING vs STING abbreviations are used inconsistently.

Answer:STING is not used in this manuscript, only STRING database is used for PPI analysis. We checked all the paper.

Result:

-         - Fig. 1 panel labeling is inconsistent with the figure legends. Fig 1D legends states that Venn plots revealed 2 up-regulated miRNAs, which again is inconsistent with the figure and text.

Answer:There were omissions in the picture during the editing process. Here we re-edited the picture so that the content of the picture is consistent with the content of the text description.

-         - Fig 2 panel labeling is inconsistent with legends.

Answer:This is a problem we had in the photo editing room, and we have corrected it

-         - The Authors state RSV-infected mice exhibited increased peribronchial and perivascular infiltration of inflammatory cells, but the results are not convincing based on only 1 representative picture of each group. It would be more informative to include at least semi-quantitative scoring for the relevant parameter(s).

Answer:We added the identification of RSV infection using real-time PCR.   

 - Fig 3D demonstrates Rn changes after metacholine challange. I would recommend to use the same range on y-axis for better representation of time-dependent Rn changes. Please do not use linear x-axis, but rather show only the proper metacholine concentrations used on the x-axis, it would be more informative.

Answer:We revised the x-axis.

-         - Why are the control values of real-time PCR analyses not 1.0 in all cases?

Answer:Control values of qRT-PCR was calculated using the actual value of the internal parameter over the mean value of the internal parameter, so the control values were not 1.0

-         - The Authors state that levels of Cxcl10, Cd14, Zeb1 and Rhoh in lung tissues were significantly increased on day 7 of RSV infection”, which is again controversial based on Fig 3F.

Answer:Yes, there is a dispute here, and in order to eliminate this dispute, we have removed Zeb1 from the statement.

-         - Please use consistent labeling for „Healthy” data on Fig 4.

Answer:Consistency changes have been made

-         - The Authors state that „The results revealed a significant upregulation of CD14 expression in all three diseases (Figure 4D)”, which looks inconsistent with the Figure, based on which in Asthma and COPD patients CD14 expression showed downregulation, although not significant.

Answer:It should be Supplementary Figure 1D. Figure 4 describled the expression of hub genes on bronchial epithlial cells. Supplementary Figure 1 describled the expression of hub genes on blood cells. CD14 is biomarker of macrophages.

-         - The Authors then elaborate on the results of the 10 hub genes in the blood of RSV-infected mice and refer to Fig 4E, which is clearly not the data the Authors refer to as.

Answer:This is an editorial error. In fact what should be quoted here is Supplemental Figure 1E, which we have corrected in the manuscript.

-         - AHR and AHD are used may times throughout the text, although AHR is not once elaborated, one can only guess it stands for airway hyperresponsiveness and used as a synonim for AHD.

Answer:AHR is airway hyper-responsiveness, and AHD is airway hyperresponsive diseases, so we redefined AHR for the first time.

Reviewer 3 Report

Introduction

Some grammar and language correction needed.

Insufficient referencing in the introduction. Article needs referencing for background information. Further, there are two statements saying “studies have shown… “ without any reference.

A short statement about whether the miRNAs you have implicated in RSV disease are encoded by the host or the virus would be useful for clarity.

Materials and methods.

Section 2.1

First 3 datasets are described as miRNA expression profiles. Does that mean that the other 5 used are just mRNA? This is unclear. Several datasets are undescribed in the text.

Microarray dataset is small-

8 datasets used

In one dataset 9 mild COPD, 20 moderate COPD.

2.11 Pentobarbital dose is unclear. Also, if it is 181 mg/kg, this is a very high dose that can affect immunological markers.

Results

3.1 Data pre-treatment is not described or referenced.

3.1 The numbers don’t add up. GSE142237 had a total of 109 ‘differentially expressed miRNAs’, with 82 up regulated and 96 down regulated. This makes a total of 178. Similarly, 10 DEmiRs in GSE33336, 2 up, 2 down = 4.

Figure legend states the figure shows “A-C, Volcano map for DEmiRs in GSE62306, GSE33336, and GSE142237”, but these data sets are described in the materials and methods as having a mix of healthy and sick individuals. It is not stated whether the up or down regulation of gene expression was in the healthy or the sick subpopulations of the miRNA datasets. This is critical for drawing any conclusions about the role these genes play during infection.

Figure 1A, X axis title typo “log2 fold change”

Figure 1B, numbers are unclear in Venn diagrams.

Figure 1C. The colour of the box plots for the healthier population and less healthy populations switches. Should keep consistent.

Figure 2A axis title, again should be log2 fold change

Figure 2C. Higher resolution image or different colour/layout needed to make text legible. Also, an axis title and a description in the figure legend of what the bars represent to anyone who is not familiar with a ‘GO analysis’. Description of BP, MF, CC and KEGG should be with the 2C section of the figure legend.

2D. Figure legends need more information. Is this for the network diagram? What do the colours mean?

2E. Where is 2E?

3.3 Reference to figure 3A about pathological changes containing syncytia. There is no visible syncytia (characterised by multinucleated cells) in that image. All microscopy images in that figure need higher resolution. If referring to specific features in a histopathology image, arrows or some kind of marker are required.

“Results showed significant red fluorescence”. This is not informative. Describe what the IFA was targeting so that the reader doesn’t have to refer to the materials and methods.

The term ‘mmu-miR-34b/c-5p’ is introduced with no description of what ‘mmu’ means.

Figure 3 figure legend needs much more detail. Reader needs to be able to interpret the data without reading the rest of the paper.

Figure 4. B and C, the writing under the graph is overlapping and unreadable. Remove or describe by a different method.

Figure 4D. logFC on x axis is not explained. Log fold change? Figure legends throughout this paper need much more detail.

Figure 5. Axis titles and microscopy are difficult to see. Better image quality needed. Figure legend needs information about the infection conditions for B and C (48 hours?).

Figure 5D was just confirming transfection efficiency? Were the cells treated or washed to remove the miRNA in solution? Some details are needed.

5E. Can’t review as I can’t read the text on the figure.

5G. If the y axis is ‘relative luciferase activity’, is it relative to the miR-NC value? If this is the case, the mean value for miR-NC luciferase activity should be at 1.0, instead of the error bar being at 1.0.

3.6. It is claimed that miR-34b/c-5p are implicated in chemotaxis but there is insufficient evidence to make this claim. What is provided is transwell migration, microscopy and cell rounding.  

Figure 6A, it is not shown or described in the legend which is treated with the miRNA. This is critical for interpreting the results.

Figure 6B. The is no way that the images provided can be used to draw the brief conclusions (with no details) made in the figure legend.

Further, the transwell method is not described at all in either the relevant results section or the materials and methods. I suggest this section is removed or that the claims are softened substantially. It is a strong claim to suggest that miR-34b/c is involved in chemotaxis, given that this miRNA has been quite extensively studied in cancer and other contexts and has never before been linked to chemotaxis. Also, transwell figure in figure 6 has no description?

Discussion.

The discussion is mostly more literature review instead of an analysis and interpretation of the results. miR-34b/c is described as regulating mucus which was ‘consistent with our results’, but no work has been done on mucus or mucus-related activity in this paper.

Again, the conclusions about chemotaxis are unjustified and there is no explanation of this ‘Transwell’ method on which so many conclusions are drawn.

Author Response

Comments and Suggestions for Authors

Although RSV is a very common virus, it can sometimes cause severe problems in low-birthweight infants with immature lung function. In this paper, the authors focused on mRNA expression, which plays an important role in the mid(early)- to late-stage of RSV infection. Furthermore, the authors attempted to obtain more specific results by showing miRNA-mRNA candidates in the three major lung diseases using human cells derived from patients and RSV-infected mouse models.

To enhance the value of this paper, we request that you supplemental explanation on  some of the results and respond to the following comments.

  1. I would like to ask about the relationship between other lung diseases, asthma and COPD, and post-RSV infection: did the specimens of the 29 patients (9 with mild COPD, 20 with moderate COPD) have any symptoms or history of RSV infection in the past few months?

Answer:The data for bioinformation analysis is derived from the GEO database. Through the analysis of the information collected from the samples in the asthma and COPD datasets, It was not introduced that the samples had symptoms or history of RSV infection in the past few months. We are collecting samples of RSV-infected asthma and COPD to determine whether miR-34b/c-5p/CXCL10 axis is the key for RSV-induced airway hyperresponsiveness.

  1. Since the severity of COPD patients is described in Materials and Methods, the severity of asthma patients should also be described. Asthma severity is described in the latest asthma guidelines published by The Global Initiative for Asthma (GINA). Also, as you are aware, treatment for asthma patients varies by age, both for adults and children. It would be more helpful if you could indicate the age of each sample or the average of the patients.

Answer:COPD data are from the GSE33336 dataset in GEO, where the severity of symptoms of COPD is described in detail. However, the asthma dataset is from GSE142237 in GEO, but there is no description of the severity of asthma symptoms in the GSE142237 dataset.

  1. You should also indicate 7 samples from asthmatics and a recent history of RSV infection; all samples from RSV patients are from children. How did you exclude samples from unrelated infections to assess mRNA expression of GSE117827, RSV and other respiratory infections may coexist. If you have information on the samples you used, please tell us how you ruled out the possibility of co-infections.

Answer:As the asthma data we used came from the GEO public datasets, the description of the samples in these data sets did not include whether the asthma patients had a history of RSV infection. However, in GSE117827, the description of the samples as multiple molecular tests performed on nasopharyngeal swabs, some viral positive results were included.

  1. Why did you decide to measure blood samples as well as tissue and mucosa samples? Also, please provide some representative publications using CD-14 to evaluate RSV infections and other respiratory diseases.

Answer: Lung tissue and mucosal are the main sites of RSV infection, and blood samples can also well reflect the global changes of the body after RSV infection. Beena Puthothu's research showed that the toll like receptor 4 (TLR4)/CD14 complex is particularly important for the initiation of an innate immune response to RSV (PMID: 17264400).CD14 is mainly expressed on macrophages, not bronchial epithelial cells.So we did the assay of the chemotaxis of macrophages to RSV-infected epithelial cells. 

  1. This paper provides a comparison of key mRNA expression in human samples and mouse models. The mouse model also allows for direct measurement of lung function and histology. The results of these experiments make this paper more informative. On the other hand, experiments using patient specimens are also very simple, so why did you add experiments using the mouse RSV infection model? Could you reiterate the significance of adding the results of the mouse RSV infection model to this paper and the limitations of this study?

Answer:However, the study still has many questions to answer. Firstly, we screened 23 target genes of miR-34b/c-5p in human studies, only a few genes have been verified at the animal level, probably because of the difference between animal and human. Secondly, because target genes of miR-34b/c-5p are distributed in different tissues, for example, CD14 is mainly distributed in macrophages and RHOH is mainly distributed in T cells, whether they are target molecules of miR-34b/c-5p remains to be further investigated. Thirdly, whether RSV induces airway hyperresponsiveness through miR-34b/c-5p/CXCL10 axis needs to be further studied by collecting clinical samples of RSV infection induced AHR. Because RSV infection is localized in epithelial cells, we mainly detected CXCL-10 expression in epithelial cells.

In Abstract: 
Please put a right parenthesis “)” just after “AHD” in the first sentence of the “Abstract: Background:”.

Answer:We have added it.

Round 2

Reviewer 3 Report

Figure 1B. 2 up regulated and 2 down regulated DEmiRs in the GSE33336 dataset = 4, not 10 as you state in 3.1.

Figure 2C and D, appear to be bar graphs with no explanation of what the lengths of the bars mean (x axis values).

Figure 3A and B show the same thing. RSV infection in the lung. 3B has no units or description of what the values mean. Are these genome copies per ml? 1000x copies per ml? The cycle threshold for detection? No information about how long the infection was allowed to proceed before quantification. Also, no primers for this PCR in the supplementary material.

Figure 6A. These images show the difference in cell morphology after treatment with PMA. There is no description of PMA in the materials and methods or what the significance of this is. Fig 6B needs some empirical quantification of the staining for it be informative.

Author Response

Figure 1B. 2 up regulated and 2 down regulated DEmiRs in the GSE33336 dataset = 4, not 10 as you state in 3.1.

Answer:Thanks for the reviewer's comments. There is indeed an editing error. We have corrected it in the manuscript and marked it in red.

Figure 2C and D, appear to be bar graphs with no explanation of what the lengths of the bars mean (x axis values).

Answer:The meaning of column length (X-axis value) is the p value of enrichment. For a more intuitive understanding, we added the trigram of GO and KEGG pathway enrichment in the supplementary materials, which are supplementary table s3 and supplementary table s4 respectively.

Figure 3A and B show the same thing. RSV infection in the lung. 3B has no units or description of what the values mean. Are these genome copies per ml? 1000x copies per ml? The cycle threshold for detection? No information about how long the infection was allowed to proceed before quantification. Also, no primers for this PCR in the supplementary material.

Answer:Figures 3A and B show the same situation, as do figures 4A and B. The result of 3B shows the relative quantification of the amount of amplification product by specific RSV primers in RSV-infected samples and non-RSV-infected samples. The cycle threshold of detection is positive when ct value is less than 35, and negative when ct value is greater than 35.

Figure 6A. These images show the difference in cell morphology after treatment with PMA. There is no description of PMA in the materials and methods or what the significance of this is. Fig 6B needs some empirical quantification of the staining for it be informative.

Answer:PMA (Phorbol 12-myristate 13-acetate), a protein kinase C inducer that induced the differentiation of monocytes into macrophages. We have added the description of the experiment in the methods section. Additional quantitative information about this result is added to the right of Figure 6B.